# Spatio-temporal assessment of annual water-balance model for Upper Ganga Basin

Anoop Kumar Shukla[1], Shray Pathak[1], Lalit Pal[1], Chandra Shekhar Prasad Ojha[1], Ana Mijic[2], Rahul Dev Garg[1]

[1]Department of Civil Engineering, Indian Institute of Technology Roorkee, Uttarakhand, India

[2]Department of Civil and Environmental Engineering, Imperial College London, London, UK

E-mail- anoopgeomatics@gmail.com, shraypathak@gmail.com, lalitpl4@gmail.com,

cspojha@gmail.com, ana.mijic@imperial.ac.uk, rdgarg@gmail.com

## Abstract

The Upper Ganga Basin, Uttarakhand, India has high hydropower potential and plays an important role in development of state economy. Thus, knowledge about water yield is of paramount importance to this region. The paper deals with use of contemporary water yield estimation models, such as the distributed model (InVEST), Lumped Zhang model and their validation to identify the most suited one for water yield estimation in this region. Earlier, while utilizing these models, attempts were made to consider a single value of some important model parameters which in fact show a variation at a pixel level scale. Therefore, in this study, the pixel level computations are performed to assess and ascertain their need in model applications. To validate the findings, the observed sub-basin discharge data is analyzed with the computed water yield for four decades, i.e. 1980, 1990, 2001 and 2015. The results obtained are in good agreement with the water yields obtained at pixel scale.

**Keywords:** Ecosystem, Evapotranspiration, Water Yield, Lumped Zhang model, InVEST model

## 1. Introduction

Accurate assessment of key ecosystem services (ES) such as water yield have gained focus in recent years in ecosystem service modelling as fresh water availability in a region are essential for agriculture, industry, human consumption, hydropower, etc. (Readhead et al., 2016). Hydrological ecosystem services often include drinking waters supply, power production, industrial use, irrigation, and many more. These hydrological ES are dependent on different factors such as watershed characteristics (e.g. topography, land use land cover (LULC), soil type) and climatic condition. To incorporate these concepts into assessment and decision making, there has been a proliferation of ecosystem modelling tools and methods. Models for ecosystem services valuation often focus on using globally available data, accepting large number of spatially explicit inputs and producing spatially explicit output, and limiting the model structure to key biophysical processes involved in land-use change (Guswa et al. 2014). Precise estimation of ES using these models is a complicated task owing to spatial variability and dependence of ES on various topographical and climatic factors. Further validation and uncertainty assessment in model output have proven to be a key obstacle to the application ES models. In the literature, studies focusing on comparison of different ES models have projected some light over the model output validation issues, however, there still exist lack of studies highlighting validation of these models for Indian basins. Further, the benefits that can be derived from ES should be analyzed and quantified in a spatially explicit manner (Sanchez et al. 2012). The uncertainties in the determination of spatial and temporal distribution of the climatic variables, especially precipitation constitutes a major obstacle to the understanding of hydrological behaviour at the catchment scales (Milly et al. 2002).

The Integrated Valuation of Ecosystem Services and Tradeoffs (InVEST) model, developed by Natural Capital Project (Tallis et al. 2010) is a tool which provides a framework to planners and

decision makers to assess trade-offs among ecosystem services and enables their comparison in
various climate and land use change scenarios. It includes a biophysical component, computing
the provision of freshwater or water yield, by different parts of the landscape and a valuation
component, representing the benefits of water provisioning to people. This model works on
simplified Budyko theory, which has a long history and still continues to receive interest in the
hydrological literature (Budyko 1979; Zhou et al. 2012; Zhang et al. 2004; Ojha et al. 2008; Zhang
et al. 2001; Donohue et al. 2012; Xu et al. 2013; Wang et al. 2014). The InVEST model applies a
one-parameter formulation of the theory in a semi-distributed way (Zhang et al. 2004). The model
is capable of quantifying water yield of a catchment under the influence of change in drivers viz.
climate variable and catchment characteristics (e.g. land use change). Various studies have been
carried out in the past demonstrating application of InVEST model. Sanchez-Canales et al. (2012)
carried out sensitivity analysis of three parameters i.e. $z$ (seasonal precipitation coefficient),
precipitation (annual) and $ET_0$ (annual reference precipitation) using the InVEST model for a
Mediterranean region basin and found precipitation to be the most sensitive parameter for the study
region. Later, Terrado et al. (2014) applied the InVEST model for the heavily humanized Llobregat
river basin. The model is applied for both extreme wet and dry conditions and the role of climatic
parameters is emphasized. Hoyer et al. (2014), applied this model in Tualatin and Yamhill basins
of northwestern Oregon under the series of urbanization and climate change scenarios.  The results
show that the climatic parameters have more sensitivity than other inputs for a water yield model.
Hamel et al. (2014), applied the same water yield model for the Cape Fear catchment, North
Carolina and concluded that the precipitation is the most influencing parameter. Goyal et al. (2017)
analyzed the InVEST water yield model for the hilly catchment by taking two catchments i.e.
Sutlej river catchment and Tungabhadra river catchment. The climate parameters i.e. precipitation
and $ET_0$ are observed to be most influencing parameters. However, there exist certain factors
limiting the application of InVEST models such as the absence or inadequate comparison with
observed data, calibration of the model without prior identification of sensitive parameters, and
lack of validation of the predictive capabilities in the context of Land Use Land Cover change (Bai
et al. 2012; Nelson et al. 2010; Su et al. 2013; Terrado et al. 2014).
The InVEST model operates on the principle of Budyko theory (Budyko, 1958, 1974). Based on
works of Schreiber (1904) and Ol'Dekop (1911), Budyko proposed formulations explaining the
relationship between precipitation and potential evapotranspiration (PET) in order to couple water
and energy balances, defined as Budyko hypothesis. Several attempts have been made to obtain
an analytical solution of the Budyko hypothesis (Schreiber, 1904; Ol'Dekop, 1911; Turc, 1954;
Mezentsev, 1955; Pike, 1964; Fu, 1981; Choudhury, 1999; Zhang et al., 2001, 2004; Porporato et
al., 2004; Yang et al., 2008; Donohue et al., 2012; Wang and Tang, 2014; Zhou et al., 2015).
Among these approaches, solutions provided by Fu (1981), called Fu's equation gained attention
as the work represented the effect of catchment properties on water balance components by
incorporating an addition parameter 'w'. Fu's equation can provide a full picture of the evaporation
mechanism at the annual timescale. Therefore, Fu's equation could be used through top-down
analysis for providing an insight into the dynamic interactions among climate, soils, and vegetation
and their controls on the annual water balance at the regional scale (Yang et al., 2007).
Considering the lack of studies on analysis and validation of ES in Indian sub-continent especially
for Himalayan catchments and to assess the applicability of various water-balance model to
Himalayan catchments, the present work attempts to compute and analyse water yield in Upper
Ganga basin using InVEST model. The work primarily considers in detail, the spatial variation of
InVEST model parameters and uses different strategies to compute water yield. Accordingly,
water yield is estimated for four years i.e. 1980, 1990, 2001 and 2015 and the most appropriate
strategy is identified. The parameters that are computed at basins level scale in previous studies
are estimated at pixel scale in order to avoid the dependence of model parameters on size of the
catchment. In addition, pixel level estimations of water yield are expected to be accurate than
output obtained using conventional approach. The term 'finer scale' in the paper represents
incorporation of spatial variations through pixel level estimation of parameters involved in
InVEST model which are otherwise taken as lumped. The work also attempts to compare the
outcomes of spatially distributed water yield model and conventionally used lumped Zhang model.
**2. Background Theory**
*2.1 Water Yield Models*
In this section, two water yield models, i.e. InVEST water yield model, which is a distributed
model and Lumped Zhang model are described.
*2.1.1 InVEST model*
The InVEST water yield model (Tallis *et al.* 2010) is designed to provide the information regarding
the changes in the ecosystem that are likely to alter the flows. It is based upon the Budyko theory
which is an empirical function that yields the ratio of actual to potential evapotranspiration
(Budyko, 1979). To describe the degree to which long-term catchment water-balances deviate
from the theoretical limits, a number of scholars have proposed one-parameter functions that can
replicate the Budyko curve (Fu 1981, Choudhury 1999, Zhang et al. 2004, Wang et al. 2014). To
observe and represent pixel-level changes to the landscape, InVEST model incorporates explicitly
the spatial variability in precipitation and PET, soil depth and vegetation. The model operates at
grid scale and acquires the inputs in the raster format into a GIS environment such as ArcGIS.
The InVEST water yield model is based on an empirical function known as the Budyko curve
(Budyko 1974). Water yield Y (x) is determined for each pixel annually for a landscape as follows:
$$Y(x) = \left(1 - \frac{AET(x)}{P(x)}\right) \times P(x) \qquad (1)$$

where, $AET(x)$ is the actual annual evapotranspiration per pixel $x$; and $P(x)$ is the annual
precipitation per pixel $x$. Actual evapotranspiration (AET) is essentially determined by climate
factors (precipitation, temperature, etc.) and mediated by catchment characteristics (vegetation
cover, soil characteristics, topography, etc.). On the other hand, potential evapotranspiration (PET)
represents the evaporating potential of the climate system prevail at a specific location and time of
year without the consideration of catchment characteristics and soil properties (Allen et al., 1998).
Several attempts have been made in past to establish relationship between AET and PET, among
which solution provided by Fu (1981) are adopted worldwide. Fu (1981) provided an analytical
solution to the Budyko hypothesis and related AET with PET by incorporating a dimensionless
parameter '$w$' which denotes the effect of catchment characteristics.
The InVEST model uses the expression of the Budyko curve proposed by Fu (1981) and Zhang *et*
*al.* (2004). The ratio of mean annual potential evapotranspiration to annual precipitation, known
as index of dryness, is expressed as:
$$\frac{AET(x)}{P(x)} = 1 + \frac{PET(x)}{P(x)} - \left[1 + \frac{PET(x)}{P(x)}\right]^{\left(\frac{1}{\omega}\right)} \qquad (2)$$

where, $PET(x)$ is the annual potential evapotranspiration per pixel $x$ (mm); and $w(x)$ is a non-
physical parameter that influences the natural climatic soil properties. The $PET(x)$ is calculated
using the following expression:
$$PET(x) = Kc(x) \times ETo(x) \qquad (3)$$

where, $ETo\,(x)$ is the annual reference evapotranspiration per pixel $x$ which is calculated based
on evapotranspiration of grass of alfalfa grown at that location shown in the equation (6). $Kc\,(x)$
is the vegetation evapotranspiration coefficient that is influenced by the change in characteristics
of land use land cover for every pixel (Allen et al. 1998). The values of $ETo\,(x)$ are adjusted by
$Kc\,(x)$ for each pixel over the land use land cover map. $w\,(x)$ is an empirial parameter and the
expression given by Donohue et al. (2012) for the InVEST model has been applied to define $\omega\,(x)$
which is as follows:
$$w\,(x) = z \times \frac{AWC\,(x)}{P\,(x)} + 1.25 \qquad\qquad (4)$$

Thus, the minimum value of the parameter $w\,(x)$ is 1.25 corresponding to bare soil where root
depth is zero (Donohue et al. 2012) . The Donohue model was developed for Australia, however,
the online documentation on InVEST model states its application globally. The parameter z is
known as seasonality factor whose values vary from 1 to 30. It represents the nature of local
precipitation and other hydrogeological parameters. The parameter $AWC\,(x)$ depicts volumetric
plant available water content which is expressed in depth (mm) which can be expressed by
following formula for each pixel $x$:
$$AWC\,(x) = Min.\,(\text{Restricting layer depth, root depth}) \times PAWC \qquad\qquad (5)$$

Root restricting layer depth is defined as the depth of the soil upto which the soil can allow the
penetration of roots and root depth is defined as the depth where 95 percent of the root biomass
occurs. Plant Available Water Content (PAWC) is generally taken as the difference between the
field capacity and wilting point. It depends upon the soil properties and can be computed by the
Soil-Plant-Air-Water (SPAW) software. PAWC is calculated using the method described by
Mckenzie et al. (2003). Modified Hargreaves method and Hargreaves method were employed for
computing the reference evapotranspiration for the study area at pixel scale.
Modified Hargreaves method
$$ET_o = 0.0013 \times 0.408 \times RA \times (T_{avg} + 17.0) \times (TD - 0.0123 \times P)^{0.76} \qquad (6)$$
where, $ET_o$ is reference evapotranspiration, $T_{avg}$ is average daily temperature ($^o$C) defined as the
average of the mean daily maximum and mean daily minimum temperature, TD ($^o$C) is the
temperature range computed as the difference between mean daily maximum and mean daily
minimum temperature, and RA is extraterrestrial radiation expressed in [MJm$^{-2}$d$^{-1}$].
Hargreaves method
$$ET_o = 0.0023 \times 0.408 \times RA \times (T_{avg} + 17.8) \times TD^{0.5} \qquad (7)$$
where, $ET_o$ is reference evapotranspiration, $T_{avg}$ is average daily temperature ($^o$C) defined as the
average of the mean daily maximum and mean daily minimum temperature, TD ($^o$C) is the
temperature range computed as the difference between mean daily maximum and mean daily
minimum temperature, and RA is extraterrestrial radiation expressed in (MJm$^{-2}$d$^{-1}$).
For computing the extraterrestrial radiation (RA), following equation is used
$$RA = \frac{24(60)}{\pi} \times G_{sc} \times d_r \times [w_s \sin(\varphi) \sin(\delta) + \cos(\varphi) \cos(\delta) \sin(w_s)] \qquad (8)$$
where, RA is extraterrestrial radiation [MJm$^{-2}$d$^{-1}$], $d_r$ is the inverse relative distance Earth-Sun, $G_{sc}$
is solar constant equals to 0.0820 MJm$^{-2}$min$^{-1}$, $w_s$ is sunset hour angle (rad), $\delta$ is solar declination
(rad) and $\varphi$ is latitude (rad).
*Determination of Seasonality factor (z) parameter*
The seasonality factor (z) parameter varies depending upon the local precipitation patterns such as
the hydrological characteristics of the area, its rainfall intensity and topography. According to the
InVEST water yield model (Tallis et al. 2010), parameter z can be computed in three different
ways. First method is suggested by Donohue *et al.* (2012), in which parameter z is expressed as
the one fifth of the number of rain events per year. Second method is suggested by Xu *et al.* (2013),
which relates $\omega(x)$ with latitude, NDVI (Normalized Difference Vegetation Index), area, etc.
Third method experiments with various selections of w (one value of w for the entire study region)
till there is a good match between observed and computed water yield. Unfortunately, this method
is not suited to a pixel based analysis as the number of pixels will be extremely large making the
method to be computationally intensive.
*2.1.2 Lumped Zhang model*
In this model all the mean values of the parameters are used as an input to compute the average
value of the water yield for the whole watershed. In this model the averaged actual transpiration,
potential evapotranspiration, w, precipitation is used as described by Zhang *et al.* (2004)
**3.  Study Area**
The Ganga river in India is ranked amongst the world's top 20 rivers in regards to the flow
discharge. The  Ganga river is segregated into three zones, viz., Upper Ganga basin, Middle Ganga
basin and Lower Ganga basin. The area choosen for the present study, i.e., Upper Ganga river
basin is situated in the northern part of India within the geographical coordinates $30^0$ 38' - $31^0$ 24'
N latitude and $78^0$ 29' - $80^0$ 22' E longitude with an area of 22,292.1 km$^2$ upto Haridwar.. The
altitude of the study area varies from 7512 m in the Himalayan terrains to 275 m in the plains.
Approximately 433 km$^2$ of entire region of the basin is under glacier landscape and 288 km$^2$ is
under fluvial landscape. About 60% of the basin is utilized for agricultural, 20% of the basin is
under the forest area, especially in the upper mountainous region. Nearly 2% of the basin is
permanently covered with snow in the mountain peaks. Most predominant soil groups found in the
region are sand, clay, loam and their compositions. In the Upper Ganga river basin, the average
annual rainfall varies from 550 to 2500 mm (Bharati et al. 2011) and a major fraction of total
annual rainfall is received during monsoon months (June-September). The geographical location
and other information of the study area Upper Ganga river basin are represented in Fig. 1.

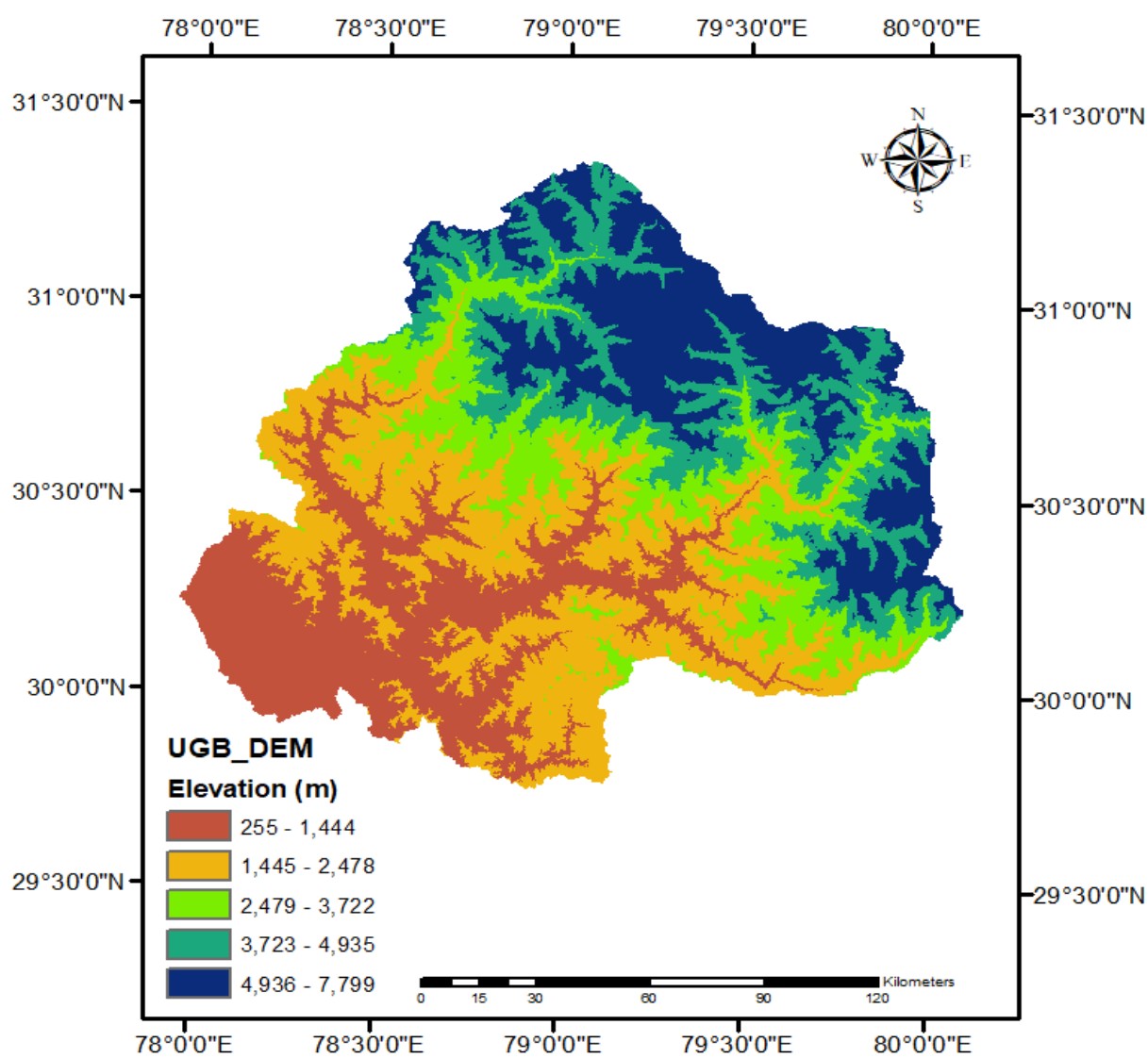


**Figure 1.** Graphical representation of study area, Upper Ganga basin

**4.  Methodology**
*4.1  Data*
*4.1.1Precipitation and Temperature*
The daily time series of precipitation and temperature for the study area is acquired from India
Meteorological Department (IMD) at a grid size of 0.25 degrees and 1 degree, respectively. The
Upper Ganga basin comes in the latitude ranging from 29.5 degrees to 31.5 degrees and longitude
ranging from 77.75 degrees to 80.25 degrees. The daily time series of precipitation was aggregated
to obtain the annual time series at each grid point. Various analysis in the study are carried out for
four years i.e. 1980, 1990, 2001 and 2015.
*4.1.2 Soil Map*
Spatial maps of soil were collected from National Bureau of soil survey and land use planning
(NBSSLUP) at 1:250000. Digital maps of soil available at a resolution of 1200m×1200m were
resampled to the resolution of land use data i.e. 30m×30m using 'resample' tool in ArcGIS in order
to maintain the scale homogeneity.  The attribute table of the raster layer contains fields like soil
depth, soil texture, percentage carbon content, drainage, slope, erosion, soil temperature and
mineralogy. The relevant feature, i.e. of soil depth and soil texture are converted into the raster
image for the Upper ganga basin.
*4.1.3 LandUse/Land Cover map*
Satellite images were acquired from different sensors of Landsat viz. Landsat 3/4 MSS/TM,
Landsat 4 TM, Landsat 7 ETM and Landsat 8 OLI sensors for the year 1980, 1990, 2001 and 2015
respectively. The images are available at different resolution and for several bands out of which
Green (G), Red (R) and Near Infrared (NIR) band images are combined to create False Colour
Composite (FCC) for the study area in ERDAS Imagine. FCCs are then classified using supervised
classification in ERDAS in six different classes, i.e. Forest, Water, Agricultural, Wasteland, Snow
and Glacier and Built-up land. Classification of the area is based upon their similar response under
different bands. Each class is then recognized with the help of ground truth and high resolution
satellite images.
### *4.2 Methodology to compute water yield*
In the present work, five different strategies are employed to compute water yield..For the ease of
presentation, these strategies are referred as A, B, C, D, E. In strategy A, an average value of
precipitation, temperature, extraterrestrial radiation and parameter 'w' is used for the entire basin.
This strategy is essentially based on Lumped Zhang Model. Strategies B, C, D and E are designated
corresponding to particular variation of InVEST model where water yield is computed using
different approach for estimating 'w' parameter. For computing parameter 'w', Xu et al. (2013)
relationship for large basin and global level is given by equation (9) and equation (10) respectively.
*For Large basins:*
$$w = 0.69387 - 0.01042 \times lat + 2.81063 \times NDVI + 0.146186 \times CTI \qquad (9)$$
*For global model:*
$$w = 3.50412 - 0.09311 \times slp - 0.03288 \times lat + 1.12312 \times NDVI - 0.00205 \times long -$$
$$0.00026 \times elev \qquad (10)$$
where, *slp* is slope gradient, *lat* is absolute latitude of basin center, *CTI* is compound topographic
index, *NDVI* is normalized difference vegetation index, *long* is longitude and *elev* is elevation.
In strategy B, entire basin is considered for computing the parameter 'w' for large basins (equation
9) by Xu et al. (2013). In strategy C, entire basin is considered for computing the parameter 'w'
for Global model (equation 10) by Xu et al. (2013). In strategy D, parameter 'w' is computed at
each pixel in order to incorporate the spatial distribution of the hydrologic variables involved in
the computations. In Strategy E, parameter 'z' is computed according to the number of rain events
in a year and subsequently equation (4) is used to compute the parameter 'w'.
For all the strategies, extraterrestrial radiation (RA) parameter is computed for each month using
equation (8) and a raster layer is generated. Precipitation data is obtained from Indian
Meteorological Department (IMD) at grid size of 0.25 degree for the study area and has been
interpreted and converted to raster format by using Inverse Distance Weighted (IDW) interpolation
technique in ArcGIS environment for obtaining the values for all pixels at a resolution equal to the
resolution of the Landsat satellite image The temperature dataset is obtained from IMD at grid size
of $1^{o} \times 1^{o}$ for the study area and has been interpreted and converted to raster format by using IDW
interpolation technique for obtaining the values for all pixels at a resolution equal to the resolution
of the Landsat satellite images. Subsequently, the mean monthly value of average temperature
(Tavg) and the difference between mean daily maximum and mean daily minimum (TD) is
obtained. The climate datasets used in the present study are of the finest resolution available so far
for the study region. The precipitation and temperature data sets were downscaled to a resolution
of land use data using Spline interpolation technique. Gridded datasets of temperature and
precipitation used in the present study has been developed using quality controlled stations and
well-proven interpolation technique. Further details about the datasets are given in Srivastava et
al. (2009) and Pai et al. (2014).

Modified Hargreaves method is applied for obtaining the values of reference evapotranspiration at each pixel for each month (Droogers et al. 2002). In this method, the inputs are $R_a$, precipitation, $T_{avg}$ and TD. Some of the months, i.e. July 1980, July 1990, August 1990, June 2001, July 2001, August 2001, June 2015, July 2015 and August 2015 showed the negative values of reference evapotranspiration as obtained from Modified Hargreaves method. Thus, for the above months the Hargreaves method as recommended by Droogers et al. (2002) is applied for obtaining the positive values for the reference evapotranspiration. Thus, all the mean values for the month are added up to get the mean yearly values for the year 1980, 1990, 2001 and 2015. To computed potential evapotranspiration, the yearly values obtained for the reference evapotranspiration have been multiplied by the vegetation evapotranspiration coefficient ($K_c$) which varies with the LULC characteristics as expressed in equation (3). The value of the vegetation evapotranspiration coefficient is taken from Allen *et al.* (1998) as shown in Table 1. In this study, Kc is taken same for all the four years from Table. 1 and is used to obtain potential evapotranspiration which is subsequently used to obtain the yearly potential evapotranspiration at each pixel of the study area.

**Table 1.** Value of $K_c$ corresponding to LandUse/LandCover classes

| S.No. | LandUse/LandCover | Percentage cover (1980) | Percentage cover (1990) | Percentage cover (2001) | Percentage cover (2015) | $K_c$ |
|-------|-------------------|------------------------|------------------------|------------------------|------------------------|-------|
| 1 | Forest | 17.84 | 16.32 | 15.78 | 15.19 | 1 |
| 2 | Water | 21.87 | 21.27 | 19.47 | 17.65 | 1 |
| 3 | Wastelands | 51.1 | 52.36 | 54.18 | 55.46 | 0.2 |
| 4 | Built-up Area | 2.07 | 2.14 | 2.27 | 2.49 | 0.4 |
| 5 | Agricultural | 3.67 | 4.04 | 3.76 | 4.22 | 0.75 |

| 6 | Snow and Glacier | 3.45 | 3.87 | 4.54 | 4.99 | 2 |
|---|---|---|---|---|---|---|

## 5. Results

### *5.1 Reference Evapotranspiration, ETo (x)*

Reference Evapotranspiration is computed for the upper Ganga Basin using a high-resolution monthly climate dataset. Modified Hargreaves method is applied for obtaining the values of reference evapotranspiration at each pixel for each month (Droogers et al. 2002). The reference evapotranspiration is a function of $R_E$, precipitation, Tavg and TD which are already computed pixel wise for each month for the year 1980, 1990, 2001 and 2015.

Some of the months i.e. July 1980, July 1990, August 1990, June 2001, July 2001, August 2001, June 2015, July 2015 and August 2015 showed negative values of reference evapotranspiration on applying Modified Hargreaves method. Thus, for the above months, the Hargreaves method is applied for obtaining the positive results. Hence, all the mean values for the months are added up to get the mean yearly values of evapotranspiration for the years 1980, 1990, 2001 and 2015, as represented in Fig 2.

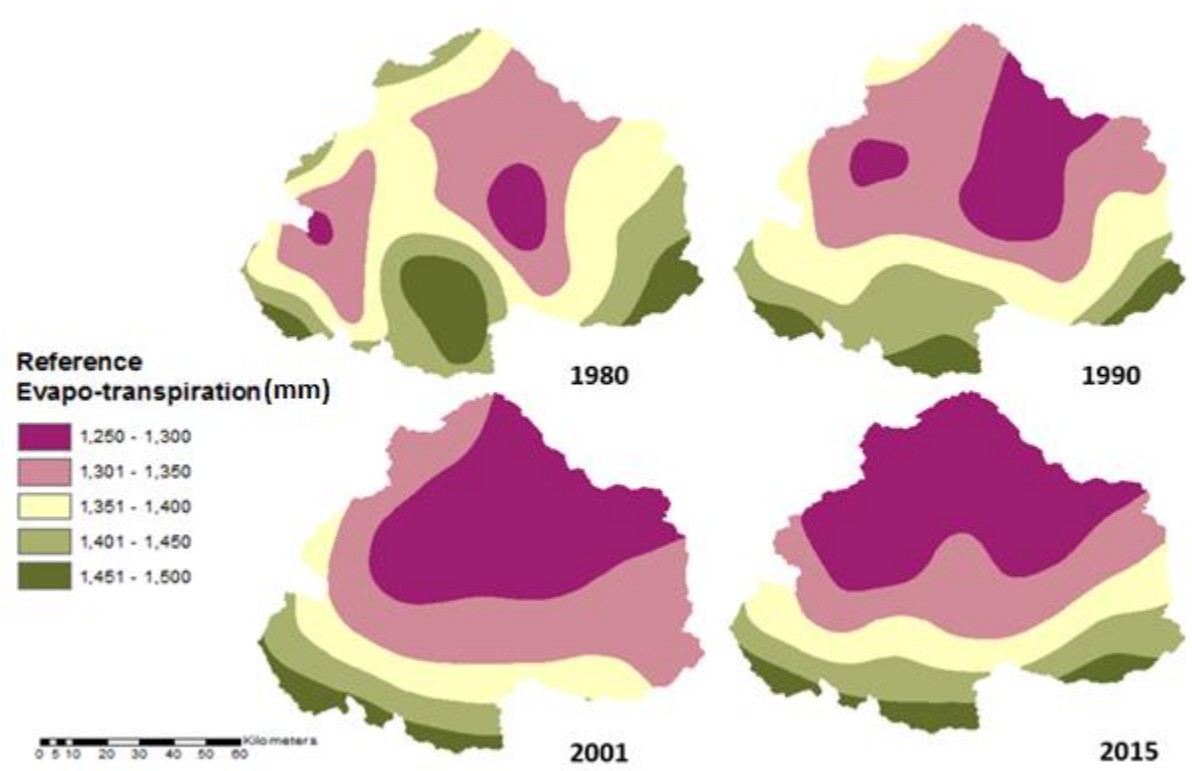

299

**Figure 2.** Reference Evapotranspiration (mm) of Upper Ganga Basin for the years 1980, 1990,

2001 and 2015.

*5.2 Potential Evapotranspiration, PET (x)*

The annual values obtained for the reference evapotranspiration is multiplied by the vegetation

evapotranspiration coefficient (Kc) which varies with the Land Use Land Cover characteristics, as

expressed in equation (3). The value of the vegetation evapotranspiration coefficient is taken from

Allen et al. (1998). The values of the vegetation evapotranspiration coefficient are taken from the

Table 1. Thus, the potential evapotranspiration is computed for Upper Ganga Basin for the years

1980, 1990, 2001 and 2015 as represented in Fig. 3.

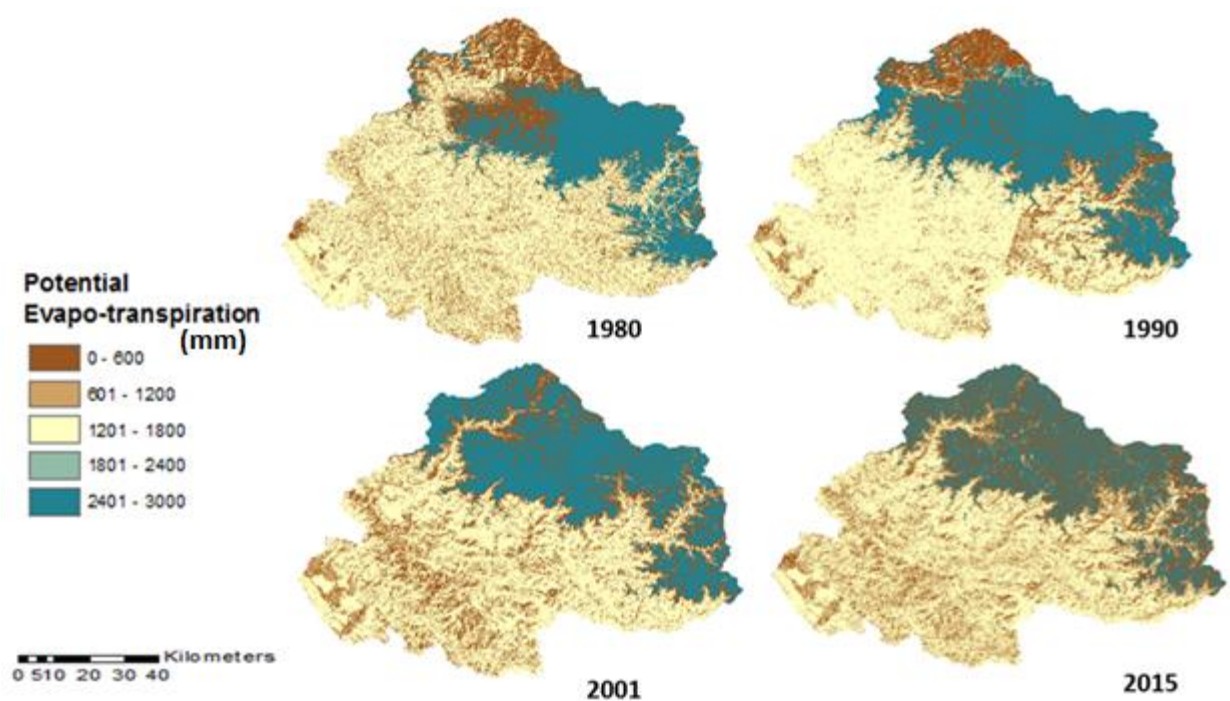

309

**Figure 3.** Potential Evapotranspiration (mm) of Upper Ganga Basin for the years 1980, 1990, 2001

and 2015.

*5.3 Water Yield, Y(x)*

As mentioned in the methodology, the water yield for the Upper Ganga basin are computed using

five strategies A, B, C, D and E:

***Strategy A: Water yield computed using Lumped Zhang Model***

Here, the basin average values of all the input parameters are considered and the water yield is

computed for the Upper Ganga basin for the year 1980, 1990, 2001 and 2015 which are obtained

as 658.52 mm, 925.68 mm, 603.71 mm and 1194.25 mm, respectively.

***Strategy B: Water yield obtained by taking the single weighted mean value of parameter 'w'***

***from Xu et al. (2013) for Large basins.***

By considering a single value of the parameter 'w' for the whole basin the water yield is computed
for Upper Ganga basin (equation 9). The weighted mean value for the parameter 'w' for the years
1980, 1990, 2001 and 2015 are obtained as 1.507, 1.541, 1.403 and 1.507 respectively. The spatial
distribution of water yield for the Upper Ganga basin for different years are represented in Fig. 4.
The mean values of water yield as obtained using this method for the year 1980, 1990, 2001 and
2015 are 755.65 mm, 959.48 mm, 742.39 mm and 1131.42 mm respectively.

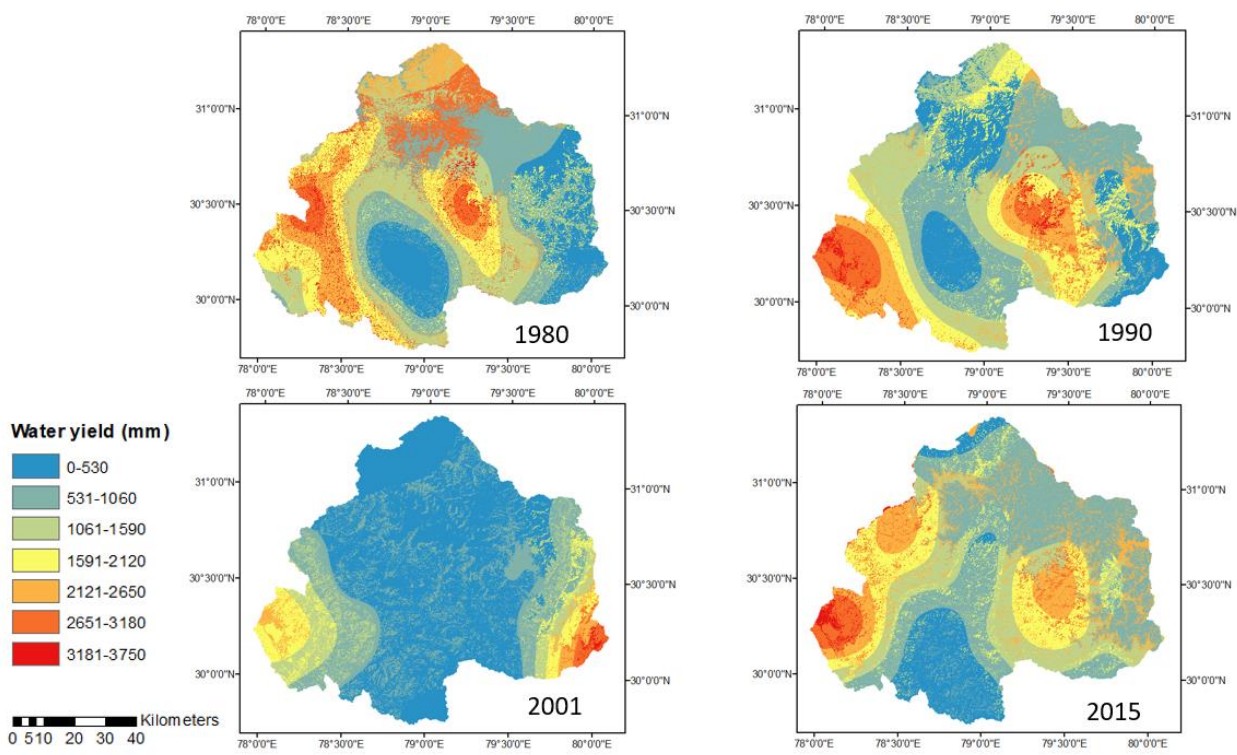


**Figure 4.** Water yield obtained by taking the single weighted mean value of parameter 'w' from
Xu et al. (2013) for large basins.
*Strategy C: Water yield obtained by taking the single weighted mean value of parameter 'w'*
*from Xu et al. (2013) for global model.*
By considering a single value of the parameter 'w' for the whole basin the water yield is computed
for Upper Ganga basin (equation 10). The weighted mean value for the parameter 'w' for the years
1980, 1990, 2001 and 2015 are obtained as -0.967, -0.955, -1.010 and -0.968 respectively. The
spatial distribution of water yield for the Upper Ganga basin for the years are shown in Fig. 5. The
mean values of water yield for the year 1980, 1990, 2001 and 2015 are 1239.92 mm, 1549.46 mm,
1149.93 mm and 1754.59 mm respectively.

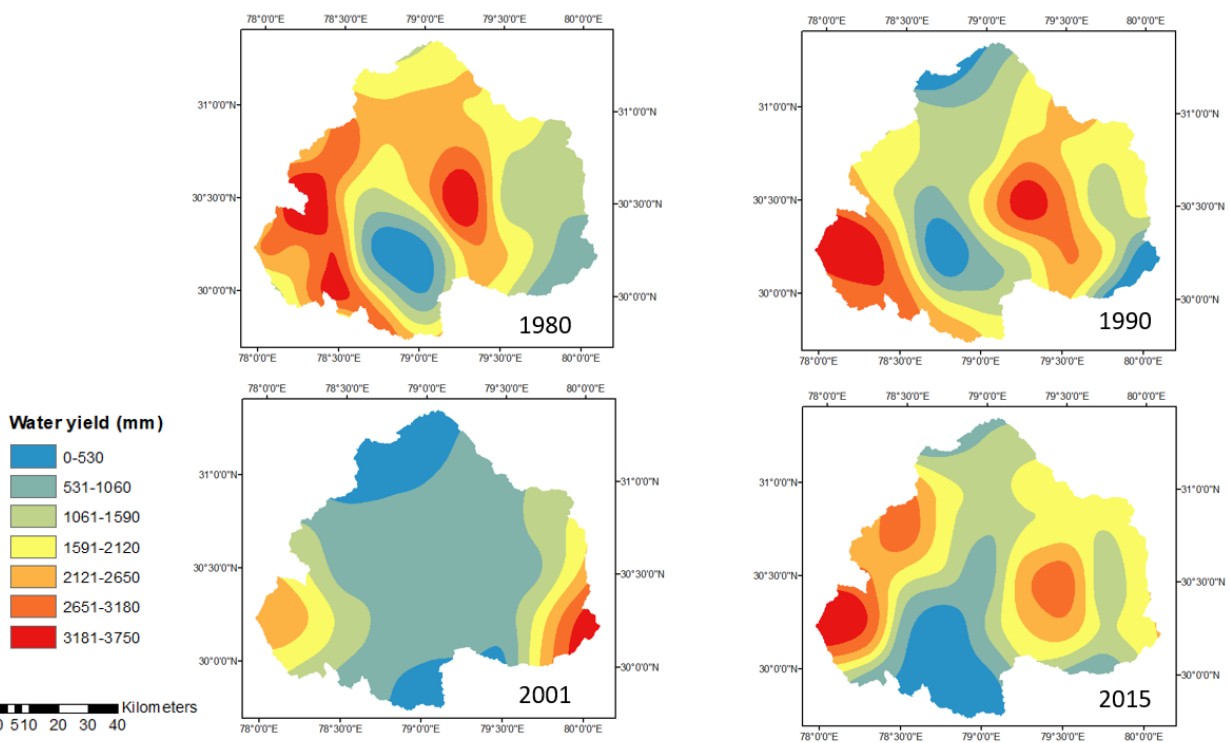


**Figure 5.** Water yield obtained by taking the single weighted mean value of parameter "w" from
Xu et al. (2013) for global model.
***Strategy D: Water yield obtained  using pixel level estimation of parameter 'w' from Xu et al.***
***(2013)***
In this strategy, the values of parameter 'w' is computed at pixel level. The water yield computed
for the years 1980, 1990, 2001 and 2015 for the Upper Ganga Basin are represented in Fig. 6. The
mean values of water yield for the year 1980, 1990, 2001 and 2015 are 1240.02 mm, 1549.44 mm,
1149.89 mm and 1754.62 mm respectively.

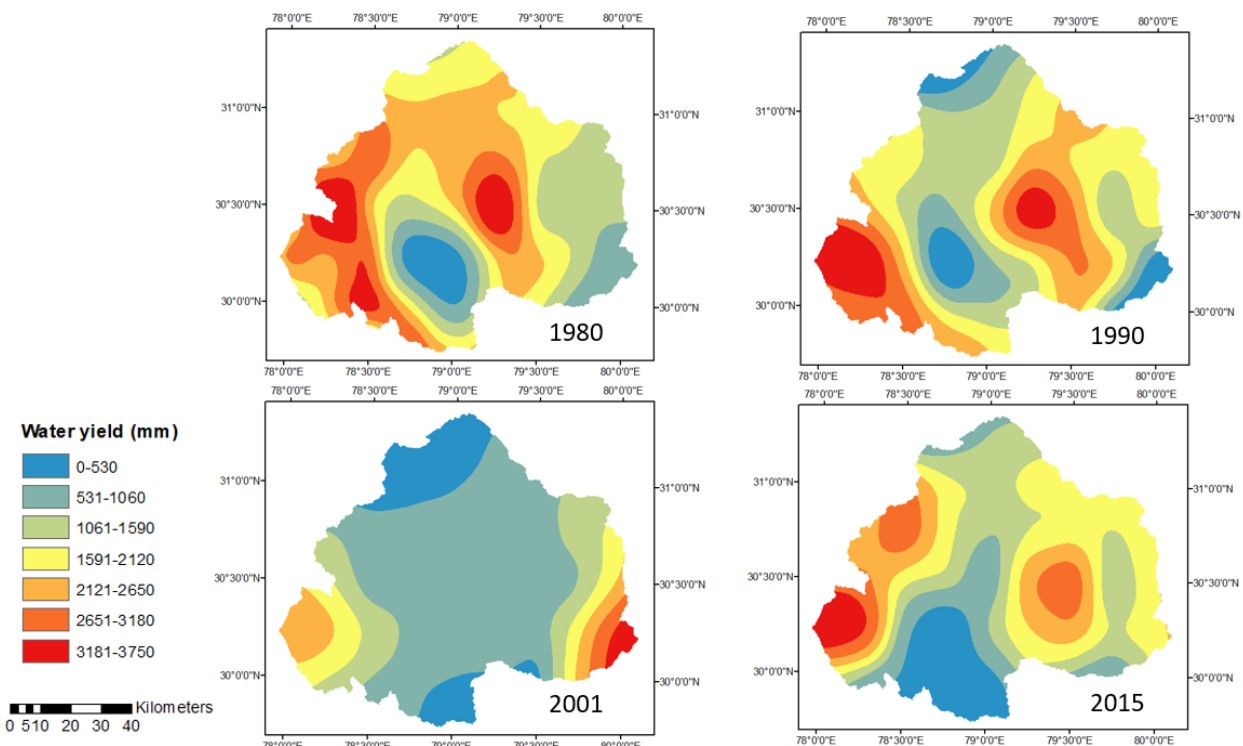


**Figure 6.** Water yield obtained by computing pixel wise value of parameter "w" from Xu et al.

(2013)

*Strategy E: Water yield obtained using pixel level estimation of parameter 'w' from Donohue et*

*al. (2012)*

The equation (4), represents the parameter 'w' which is a function of the parameters 'z', AWC and

P. The parameter 'w' in the equation involved in strategy 'E' have been proposed by Donohue et

al. (2012) which is also cited in online documentation of InVEST model, however, the final

equation used for estimating water yield is from the InVEST model. Considering this fact,

Donohue et al. (2012) has been cited in Strategy 'E'. The water yield is computed for Upper Ganga

Basin for the years are shown in Fig. 7. The mean values of water yield for the years 1980, 1990,

2001 and 2015 are 1241.09 mm, 1552.38 mm, 1153.95 mm and 1753.53 mm respectively.

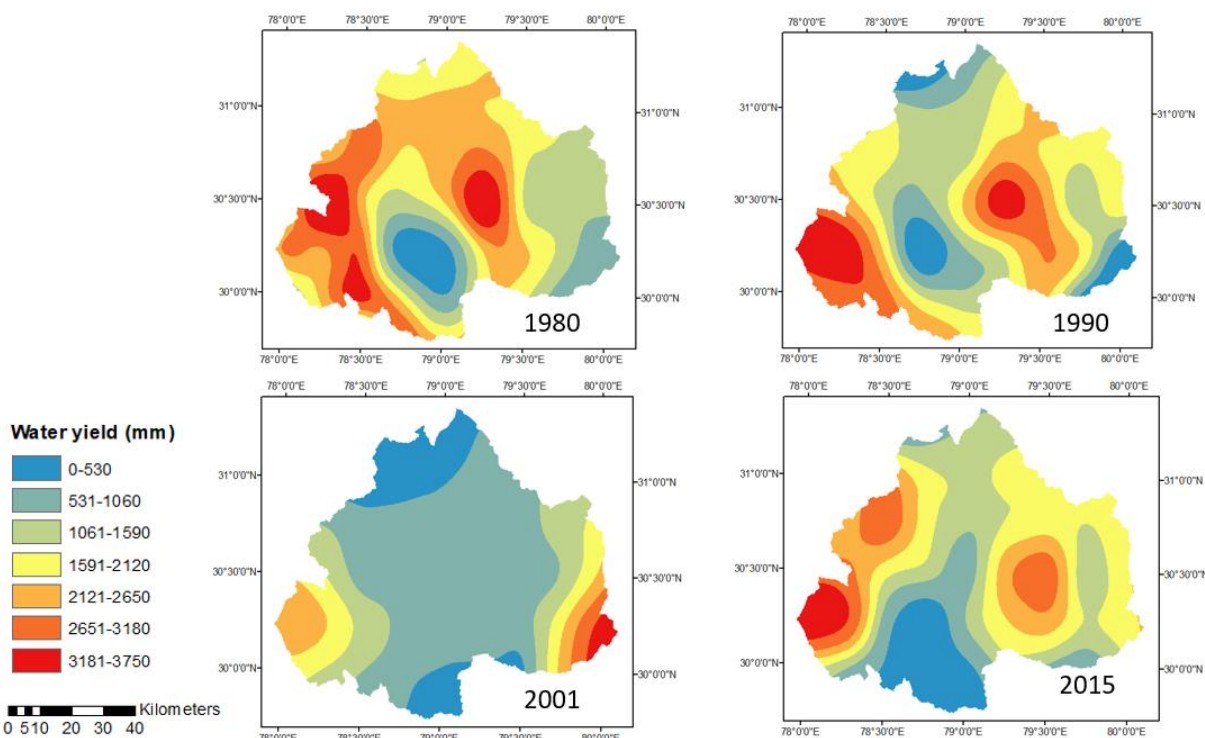


**Figure 7.** Water yield obtained by computing pixel wise value of parameter "w" from Donohue *et*
*al.* (2012)

### *5.2 Validation of ET and water yield estimates*

For validation purpose, the basin average annual values of PET and AET estimated using various
strategies are compared with the corresponding basin average values obtained from available
global datasets (Table 2). Model simulated AET values are obtained from GLDAS global ET
datasets from Noah model outputs. Basin average values of PET dataset are obtained from Climate
Research Unit (CRU) PET datasets (CRU TS v. 4.01) available at resolution of $0.5^{\circ}$. From the
comparison, both AET (GLDAS) and PET (CRU TS) values are found to in agreement with the
satellite estimated values. Spatial of Global datasets of AET and PET are shown in Figure 8 and
9, respectively.

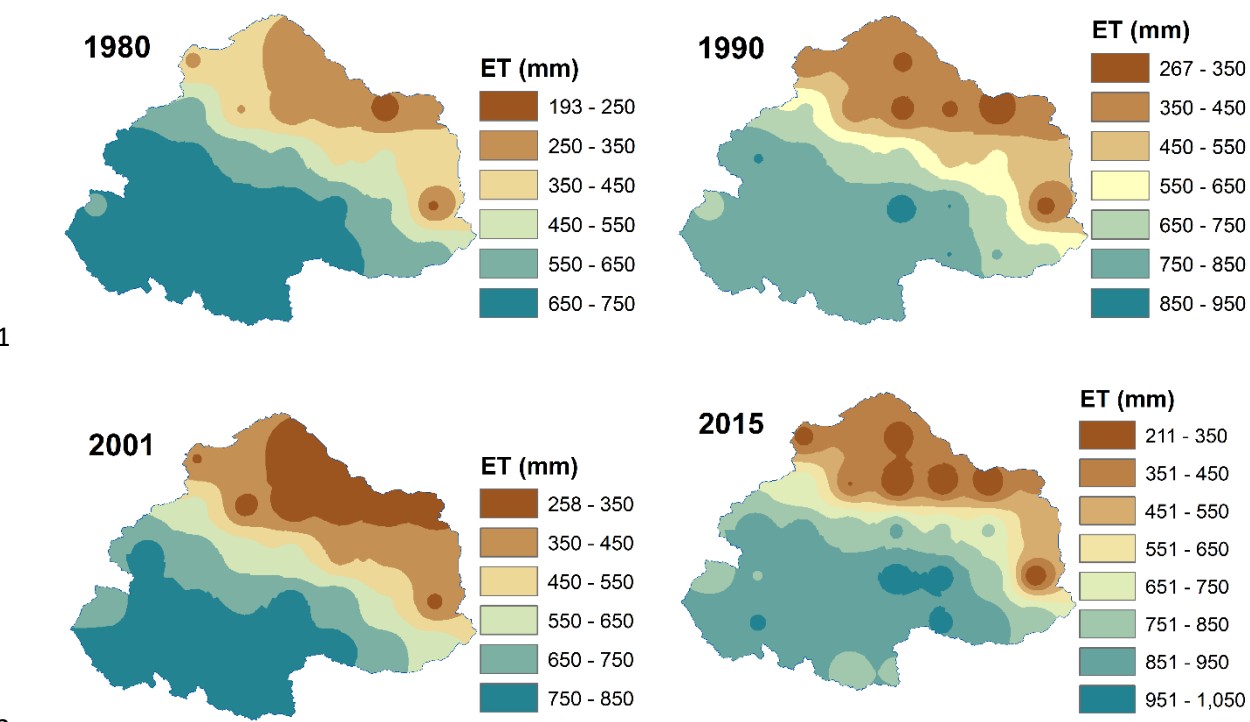


**Figure 8.** Spatial distribution of AET obtained from GLDAS Noah output datasets.


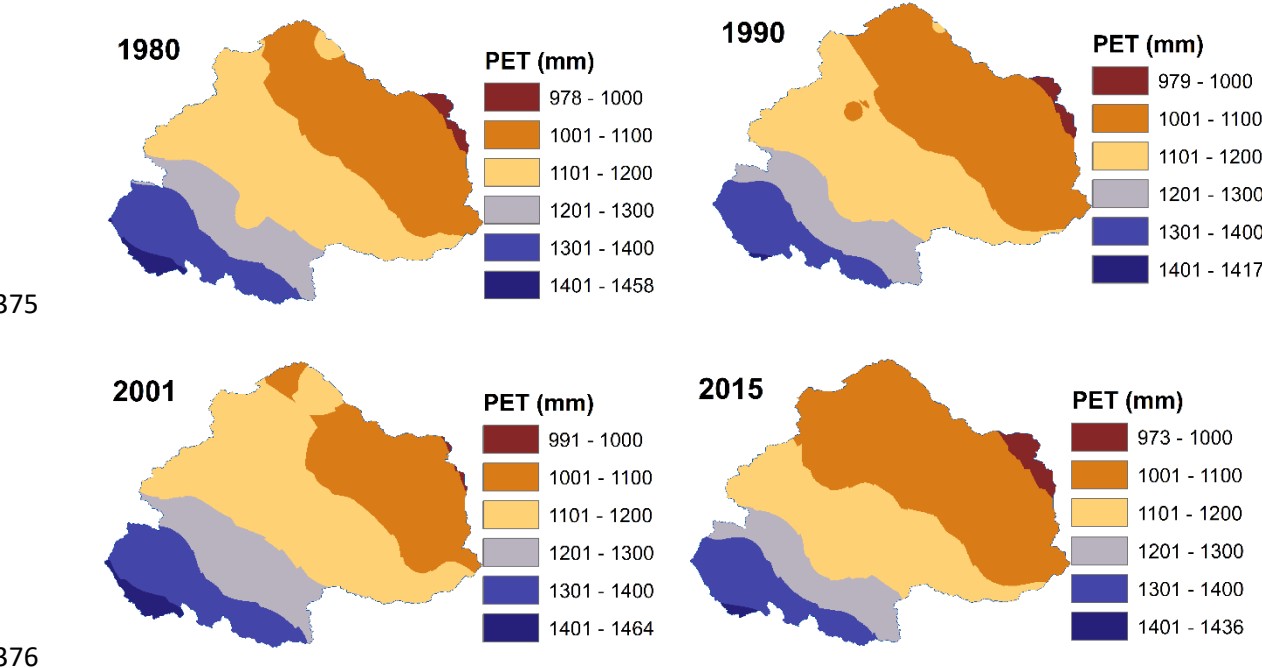


**Figure 9.** Spatial distribution of PET obtained from CRU datasets.


Table 2: Comparison of model estimated PET and AET with satellite estimates

| Parameter | | | | | InVEST model | | | |
|---|---|---|---|---|---|---|---|---|
| (mm) | Year | Source 2 (GLDAS) | Source 2 (CRU) | Strategy A (Lumped Zhang Model) | Strategy B (Large Model) | Strategy C (Global model) | Strategy D (Xu et al. 2013) | Strategy E (Donohue et al. 2012) |
| AET | 1980 | 555.0355 | | 696.84 | 486.07 | 679.52 | 679.68 | 680.01 |
| | 1990 | 646.168 | | 815.02 | 592.3 | 735.23 | 735.27 | 736.25 |
| | 2001 | 588.084 | | 680.76 | 408.86 | 548.28 | 548.39 | 550.38 |
| | 2015 | 716.8316 | | 900.11 | 625.41 | 743.48 | 743.52 | 744.34 |
| | | | | | | | | |
| PET | 1980 | | 1175.964 | 1376.64 | 1382.12 | 1382.12 | 1382.12 | 1382.12 |
| | 1990 | | 1156.497 | 1456.16 | 1461.86 | 1461.86 | 1461.86 | 1461.86 |
| | 2001 | | 1184.847 | 1457.08 | 1462.96 | 1462.96 | 1462.96 | 1462.96 |
| | 2015 | | 1156.686 | 1544.20 | 1550.42 | 1550.42 | 1550.42 | 1550.42 |

The validation of the water yield obtained from various strategies is performed upto Rishikesh gauging site of Upper Ganga basin (Fig. 10). The discharge data of the basin is obtained from Irrigation department of Uttarakhand state. Present work considers runoff from both precipitation as well as snowfall for the region, but 32% of the observed discharge has been removed as it is contributed by glacier ice melt to the streamflow for this catchment as explained by Maurya et al. (2011) for our study area. The above mentioned fraction of discharge had been quantified using isotope study which separates snow melt contribution from that of the glacier melt (Maurya et al., 2011). A comprehensive work on water balance of Upper Ganga Basin has been discussed by Jain et al. (2017) (Table 4 in Jain et al., 2017). For a precipitation value of 1236.1 mm, ground water contributes by an amount of 293.92 mm and snow melt contributes by 73.84 mm. It is apprehended that ground water flow and snow melt equals to 367.76 mm which is approximately equals to 29.75 percent of Precipitation. Subsequently, this percentage contribution is also supported by the value

reported by Maurya et al. (2011). A comparison of the water yield computed and observed for the
study region for different years by various proposed strategies are shown in Table 3.

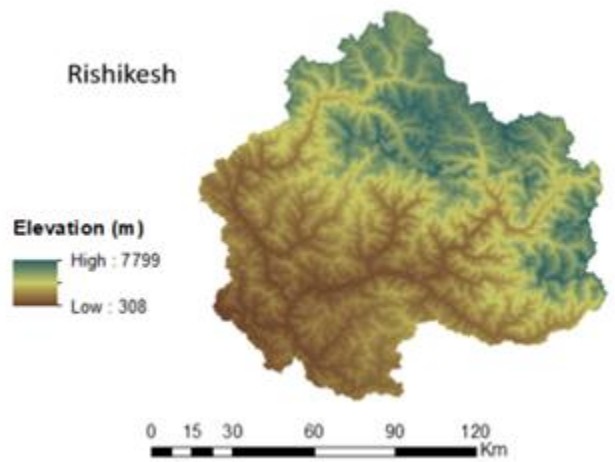


**Figure 10.** Graphical representation of sub-basin Rishikesh
**Table 3.** Observed vs computed water yield by various proposed strategies for Rishikesh sub-
basin.

| Strategies | 1980 | 1990 | 2001 | 2015 |
|---|---|---|---|---|
| Observed discharge (mm) | 1831.31 | 2422.43 | 2187.22 | 2835.81 |
| Observed (mm) (after reducing approx. 32% snow melting contribution) | 1245.29 | 1647.25 | 1487.31 | 1928.35 |
| Water Yield_Strategy A (mm) | 652.47 | 914.35 | 598.25 | 1189.72 |
| Water Yield_Strategy B (mm) | 745.38 | 917.77 | 697.75 | 1092.17 |
| Water Yield_Strategy C (mm) | 1229.90 | 1506.82 | 1102.62 | 1718.17 |
| Water Yield_Strategy D (mm) | 1229.99 | 1506.74 | 1102.61 | 1718.18 |
| Water Yield_Strategy E (mm) | 1230.77 | 1508.88 | 1106.86 | 1720.16 |


Values of water yield estimated using strategy A to E are systematically increasing but are not
steady in nature as water yield estimated using strategy A and B lies in range 650 – 750 mm
whereas water yield from strategy C-E lies in range 1229 – 1231 mm for the year 1980 (see Table
3). Similar results are also evident for other years too. Also, water yield estimated using strategy
C-E are more or less same for a given year as these strategies involve pixel based estimation of
water yield considering spatial variation in Budyko parameters. Parameters involved in Budyko
model such as 'w' are found to be dependent on various factors such as catchment characteristics,
vegetation cover, etc. as well as climate seasonality (Li et al. 2013). Ahn and Merwade (2017)
have analysed the relationship between basin characteristics and factor 'w' for 175 stations spread
over the USA results are presented in Ahn and Merwade, (2017). As evident from their study, no
precise conclusion can be drawn regarding relationship between basin characteristics and value of
'w' especially in case of basin area characteristics. Moreover, no straight forward relationship has
yet been identified between basin characteristics and model parameters and it is a subject matter
for further study.
6. **Discussion**
The study aimed to apply the InVEST water yield model, a tool that is gaining interest in ecosystem
services community for Upper Ganga Basin, having the variability in the topography and
consisting of hilly areas, plain areas and the regions which are totally covered with snow. The
InVEST model is based upon Budyko theory which requires low amount of data and low level of
expertise, thus making it acceptable world-wide. Monthly precipitation, monthly average value of
temperature, monthly value of difference of mean daily maximum and mean daily minimum and
extraterrestrial radiation parameters are computed for the Upper Ganga Basin for each month of
all the four years i.e. 1980, 1990, 2001 and 2015 and converted into the raster format for the further
analysis. The monthly reference evapotranspiration is thus computed using input parameters in the
GIS environment by applying the modified Hargreaves equation for all the months except some
months where the modified Hargreaves equation shows the negative results for the reference
evapotranspiration value. For those months Hargreaves method is applied to obtain the positive
value of reference evapotranspiration as also suggested by Goyal et al. (2017). Reference
evapotranspiration when multiplied with $K_c$ gives the potential evapotranspiration. All the monthly
values of different years are added up to obtain the yearly value of reference evapotranspiration.
$K_c$ is the function of Land Use Land Cover, thus supervised classification is done to prepare the
raster Land Use Land Cover map for the Upper Ganga Basin. Thus, the yearly value of potential
evapotranspiration is obtained for the study area for the years 1980, 1990, 2001 and 2015.
The paper focuses on all the methodologies discussed in the paper and is applied on the Upper
Ganga basin. Thus, water yield is computed both from InVEST model as well as Lumped Zhang
model. The value of the parameter 'w' are computed in four ways, i.e. mean single value obtained
from Xu et al. (2013) for large basins and global model, pixel wise value of parameter 'w' from
Xu et al. (2013) and pixel wise value of parameter 'w' from Donohue et al. (2012). Although, the
Upper Ganga basin lies in large basin category as per the definition from Xu et al. (2013), but, the
yield computed using global model is in good agreement with the observed data for the Upper
Ganga basin. In the study, pixel level estimation of parameter 'w' is made in order to incorporate
the spatial variability of the parameter in water yield estimation. Thus, two pixel wise values of
parameter 'w' is computed for the Upper Ganga basin for years 1980, 1990, 2001 and 2015 by
considering two approaches as given by Xu et al. (2013) and Donohue et al. (2012). Also, the water
yield is computed from Lumped Zhang model which works on the approach of considering mean
values of all the parameters indulged in the computations of water yield. Thus, in five ways water
yield are computed for the Upper Ganga basin for the years 1980, 1990, 2001 and 2015.
At Rishikesh gauging site,surface runoff data is obtained by extracting the snow melt from the
discharge data as the snow melting contributes about 32 percent of total runoff (Maurya et al.,
2011). Using this fact, the observed yield is compared with the computed water yield based on
different proposed strategies for the years 1980, 1990, 2001 and 2015 represented in Table 3. The
results obtained from Donohue et al. (2012) and Xu et al. (2013) computed at pixel level (Strategy
C, Strategy D and Strategy E), thus represents better performance than other and are in good
agreement with the observed data. It is clear that in order to go for hydrological processing for any
watershed, pixel wise computation is advisable.  The parameters involved in the Budyko model
are dependent on various factors such as basin characteristics (size, topography, stream length,
slope, etc.), climate seasonality, etc. (Li et al., 2013). The factors affecting model parameters again
vary both spatially and temporally. Moreover, the relationship between these factors and model
parameters are not yet well defined (Ahn and Merwade, 2017). In such scenario, adopting a
hypothesis by assuming few of these controlling factors (such as 'w') to be constant spatially or
temporally is inappropriate. Considering these facts, the present study attempts to incorporate the
spatial variability of model parameter for estimation of water yield at pixel level. As the
computations are made at pixel level in GIS environment, the assumption of dependence of model
parameters over scale of the catchment may also be disregarded. The computations made in present
work are based on empirical equations, however, the application of these equations has been well
documented worldwide for estimation of various water balance components at various basin scales
(Zhang et al., 2008; Ma et al., 2008; Ning et al., 2017; Rouholahnejad et al., 2017; Wang et al.,
2017). Hence, it is recommended, that for such a large basin there is a strong need to compute all
the parameters involved in the computations of water yield at pixel scale rather than adopting the
mean values for entire watershed.
**7.   Summary and Conclusions**
The present study aimed to apply the InVEST annual water yield model, a tool that is gaining
interest in the ecosystem services community. While such simple models having low requirements
for data, high level of expertise are needed for practical applications of such model as a single
representative value of model parameter for the entire basin does not provide good estimates of
water yield. On the other hand, performing pixel scale computation of water yield indicates a better
performance and results obtained show better agreement with the observed water yield. As far as
parameter 'w' is concerned, global model works better than other representation of 'w' available
in literature. The water yield is computed using five different strategies and results are analyzed
with the observed data of sub-basins of Upper Ganga Basin. The present study attempts to quantify
annual water yield at pixel level irrespective of the size of catchment. Therefore, the proposed
methodology is expected to perform well for the catchment of any given size. Changes in
catchment's water storage over time are required to be quantified in order to validate the
applicability of Budyko's model to long term data for the catchment under study.  Earlier, some
of the important parameters for the water yield used to be computed at a basin level scale which
brings noise in the results. Thus, by considering all the parameters involved in the model at pixel
level scale, the results obtained are higher in accuracy.
The study attempts to incorporate the spatial variability of parameters involved in the model
thorough pixel level estimation of parameters which are otherwise taken as lumped in the previous
studies. Study results show that the water yield estimated considering spatial variability in model
parameters are in better agreement with the observed water yield as compared to the water yield
estimated by considering the parameters to be lumped over the study region. Further, the
computations of various parameters are made at pixel level, therefore, the estimates of water
balance components using this approach are expected to be independent of the assumption of
dependence of parameters on catchment size. As the variation between Budyko's model
parameters and their controlling factors has not shown well defined relationship (Ahn and
Merwade, 2017), the study emphasizes water yield estimation using pixel based computations.
Thus it can inferred that: (i)between two approaches used, i.e. considering entire basin and pixel
level approach, the pixel level approach is found to provide better results and (ii) in pixel level
based computations, results further improved with the use of a parameter 'w' based on a global
model than regional models of 'w' for large basins in Himalayan basin.
**Acknowledgement**
Authors are thankful to Executive Engineer, Irrigation Department, Uttarakhand, for providing the
discharge data for the Rishikesh sub-basin of Upper Ganga Basin.

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
