# Peer review of "Spatio-temporal assessment of annual water-balance model for Upper Ganga"

_Hydrology and Earth System Sciences, 2017_

## Referee Comment (RC1) · Anonymous Referee #1 · 26 Oct 2017

The manuscript generates component of annual water balance based on a hydrologic model. The analysis is very routine and there is very little validation of output. Some of the concepts also needs to be corrected such as, the ET is calculated without considering wind and humidity. The manuscript, in this form, is not suitable to be published in HESS.

1. There is no validation for variables such as ET and soil moisture. The authors must validate the model with satellite estimates of ET and soil moisture.

2. There is no specific scientific hypothesis, the article just reports results from some empirical equations without proper analysis.

3. I do not see a proper conclusion coming out of this work.

[Figure]

4. The write up is extremely poor and needs significant revision.

---

## Referee Comment (RC2) · Anonymous Referee #2 · 8 Nov 2017

**Spatio-temporal assessment of annual water-balance model for Upper Ganga Basin By Shukla et al.**

This study estimates the water yield for the Upper Ganga basin using variations of the Budyko model that relates aridity index (ratio of long term potential evapotranspiration to precipitation) to evaporation ratio (ratio of long term actual evapotranspiration to precipitation). Several versions of a parameterized form of the Budyko curve, developed over the past years, are used to estimate long term streamflow for the basin. It is also assessed whether the inclusion of spatial variability of basin properties improves predictive skill. Though the study does not make any new contributions, it has the potential to contribute to understanding hydrology of this particular basin. However, in its present form it has major drawbacks:

1. Literature review: The study overlooks a significant body of literature in streamflow modeling in the region. Studies are available both at the scale of entire India, the Ganga basin as well as finer scales. In addition, the premise of the study is poorly developed and developments related to Budyko's theory are improperly explained. In fact, the work by Donohue et al. (2012) is cited but equations from InVEST's online documentation are instead used. It is not straightforward to connect the equations in the manuscript with Donohue et al. (2012) formulations. Overall, the introduction needs connection to a wider literature base, along with better exposition of developments in Budyko theory.

2. Methods: The methods rely on previously developed relationships between Budyko parameter and observable catchment properties. However, some of these relationships, such as those in Donohue et al. 2012 were developed for Australia. Similarly, Xu et al. (2013) report that the global model could explain only 53% of observed variation of Budyko's parameter in their dataset. The large basin model worked well but is the Upper Ganga basin large enough in comparison to the 32 basins used in Xu et al. (2013)?

3. Climate data: The resolution of climate data used to compute fine scale variables is of concern. The introduction stresses on a stronger control of precipitation (and potential evapotranspiration) on runoff estimates, as compared to Budyko's parameter, but the analysis works with coarse climatic data. Though precipitation and temperature data were downscaled to the resolution of land use data (by a statistical technique that is not described well.), the effect of elevation on these variables was neglected (for example lapse rate was not accounted for in temperature estimates). As the basin has significant elevation variations, this may lead to biases in water yield estimates.

4. Validation: For the validation catchment, 32% of observed discharge is removed as it is assumed to be snowmelt. But snow melt still counts within the hydrological budget of the region as it is contributed by precipitation falling as snow, which is being used in the Budyko model. If the melt contribution was from long term glacier melts that contribute water to the region in addition to precipitation falling as rain or snow, one may remove it. Even that will be challenging at annual time scales if the basin has significant storage. Unless the distinction between glacier and snow melt is made, and some reasoning as to why Budyko's approach can be applied at annual time steps, it will be hard to justify this reduction. There is also the issue of claiming predictive skill over an entire basin by looking at performance at a single sub-basin in a single year. Note that most approaches based on the Budyko's curve must work with long term data as even at annual time scales, catchment's water storage

changes may be significant and the Budyko model may be invalid (Donohue et al. 2007). The discussion should reflect the limitations of this approach.

5. Interpretation of results: For some reason, as we move from strategy A to E, catchment water yield steadily increases, or, ET decreases. This indicates a systematic change in the Budyko parameter as we go from simpler to more complex relationships requiring more data. Why would the Budyko parameter scale in this manner? This also seems to be in contradiction of the result by Choudhary (1999) who showed that as larger areas are used in a lumped form, Budyo's parameter changes such that actual evapotranspiration reduces. See also the discussion in Donohue et al. (2007). Given the limited data for validation, it is important to physically interpret the results instead of focusing on which method is the best.

**Minor Comment**

Structure: The paper can be re-structured to improve clarity. Sections 2 and 4 have overlapping items, while 'data' generally goes better with 'Study area'.

**References**

Choudhury, B., 1999. Evaluation of an empirical equation for annual evaporation using field observations and results from a biophysical model. *Journal of Hydrology*, *216*(1), pp.99-110.

Donohue, R.J., Roderick, M.L. and McVicar, T.R., 2006. On the importance of including vegetation dynamics in Budyko? s hydrological model. *Hydrology and Earth System Sciences Discussions*, *3*(4), pp.1517-1551.

Donohue, R.J., Roderick, M.L. and McVicar, T.R., 2012. Roots, storms and soil pores: Incorporating key ecohydrological processes into Budyko's hydrological model. Journal of Hydrology, 436, pp.35-50.

Xu, X., Liu, W., Scanlon, B.R., Zhang, L. and Pan, M., 2013. Local and global factors controlling water-energy balances within the Budyko framework. *Geophysical Research Letters*, *40*(23), pp.6123-6129.

---

## Author Comment (AC1) · 6 Dec 2017

**The manuscript generates component of annual water balance based on a hydrologic model. The analysis is very routine and there is very little validation of output. Some of the concepts also needs to be corrected such as, the ET is calculated without considering wind and humidity. The manuscript, in this form, is not suitable to be published in HESS.**

1. **There is no validation for variables such as ET and soil mositure. The authors must validate the model with satellite estimates of ET and soil moisture.**

**Reply:** As suggested by reviewers, estimated values of both ET and PET have been validated with available satellite estimates from GLDAS and MODIS (ET) and CRU TS (PET). The final equation used for estimating water yield involves two ET estimates viz. AET and PET which both are been validated using satellite based estimates for the respective years.

| Parameter | | | | | InVEST model | | | |
|---|---|---|---|---|---|---|---|---|
| (mm) | | Source 2 (GLDAS) | Source 2 (CRU) | Strategy A (Lumped Zhang Model) | Strategy B (Large Model) | Strategy C (Global model) | Strategy D (Xu et al. 2013) | Strategy E (Donohue et al. 2012) |
| AET | 1980 | 555.0355 | | 696.84 | 486.07 | 679.52 | 679.68 | 680.01 |
| | 1990 | 646.168 | | 815.02 | 592.3 | 735.23 | 735.27 | 736.25 |
| | 2001 | 588.084 | | 680.76 | 408.86 | 548.28 | 548.39 | 550.38 |
| | 2015 | 716.8316 | | 900.11 | 625.41 | 743.48 | 743.52 | 744.34 |
| | | | | | | | | |
| PET | 1980 | | 1175.964 | 1376.64 | 1382.12 | 1382.12 | 1382.12 | 1382.12 |
| | 1990 | | 1156.497 | 1456.16 | 1461.86 | 1461.86 | 1461.86 | 1461.86 |
| | 2001 | | 1184.847 | 1457.08 | 1462.96 | 1462.96 | 1462.96 | 1462.96 |
| | 2015 | | 1156.686 | 1544.20 | 1550.42 | 1550.42 | 1550.42 | 1550.42 |

**2. There is no specific scientific hypothesis, the article just reports results from some empirical equations without proper analysis.**

**Reply:** Authors agree that the study lacks a precise scientific hypothesis. However, the parameters involved in the Budyko model are dependent on various factors such as basin characteristics (size, topography, stream length, slope, etc.), climate seasonality, etc. (Li et al. 2013). The factors affecting model parameters again vary both spatially and temporally. Moreover, the relationship between these factors and model parameters are not yet well defined (Ahn and Merwade, 2017). In such scenario, adopting a hypothesis by assuming few of these controlling factors (such as 'w') to be constant spatially or temporally is inappropriate. Considering these facts, the present study attempts to incorporate the spatial variability of model parameter for estimation of water yield at pixel level. As the computations are made at pixel level in GIS environment, the assumption of dependence of model parameters over scale of the catchment may also be disregarded.

Authors also agree that the computations made in present work are based on empirical equations, however, the application of these equations has been well documented worldwide for estimation

of various water balance components at various basin scales (Zhang et al. 2008; Ma et al. 2008; Ning et al. 2017; Rouholahnejad et al. 2017; Wang et al. 2017). An illustrative summary of such studies has been added in the revised manuscript.

**3. I do not see a proper conclusion coming out of this work.**

**Reply:** Present study attempts to compute water yield from a Himalayan catchment using InVEST water yield model. The study attempts to incorporate the spatial variability of parameters involved in the model thorough pixel level estimation of parameters which are otherwise taken as lumped in the previous studies. Study results show that the water yield estimated considering spatial variability in model parameters are in better agreement with the observed water yield as compared to the water yield estimated by considering the parameters to be lumped over the study region. Further, the computations of various parameters are made at pixel level, therefore, the estimates of water balance components using this approach are expected to be independent of the assumption of dependence of parameters on catchment size. As the variation between Budyko's model parameters and their controlling factors has not shown well defined trend (see Fig 1), the study emphasizes water yield estimation using pixel based computations.

[Figure]

[Figure]

Figure 1: The relationship between basin characteristics and optimal w values (Source: Ahn and Merwade, 2017)

**4. The write up is extremely poor and needs significant revision.**

**Reply:** As per reviewer's suggestion, the write up has been improved wherever required. Our endeavor will be that the revised paper is much better than the current version.

**References:**

1. Ahn, K. H., and Merwade, V. (2017). "The Integrated Impact of Basin Characteristics on Changes in Hydrological Variables", Book Chapter 12 in "*Sustainable Water Resources Management*", American Society of Civil Engineers (ASCE), pp. 317-336. ISBN: 978-0-7844-1476-7.

2. Choudhury, B. J. (1999). "Evaluation of an empirical equation for annual evaporation using field observations and results from a biophysical model." *J. Hydrol.*, 216(1–2), 99–110.

3. Li, D., Pan, M., Cong, Z., Zhang, L., and Wood, E. (2013). "Vegetation control on water and energy balance within the Budyko framework." *Water Resources Research*, 49(2), 969-976.

4. Ma, Z. M., S. Z. Kang, L. Zhang, L. Tong, and X. L. Su (2008). "Analysis of impacts of climate variability and human activity on streamflow for a river basin in arid region of northwest China." *J. Hydrol.*, 352(3–4), 239–249.

5. Ning, T., Li, Z., and Liu, W. (2017). "Vegetation dynamics and climate seasonality jointly control the interannual catchment water balance in the Loess Plateau under the Budyko framework." *Hydrol. Earth Syst. Sci.*, 21, 1515-1526

6. Rouholahnejad Freund, E. and Kirchner, J. W. (2017). "A Budyko framework for estimating how spatial heterogeneity and lateral moisture redistribution affect average evapotranspiration rates as seen from the atmosphere." *Hydrol. Earth Syst. Sci.*, 21, 217-233

7. Wang, X.-S. and Zhou, Y. (2017). "Shift of annual water balance in the Budyko space for catchments with groundwater-dependent evapotranspiration." *Hydrol. Earth Syst. Sci.*, 20, 3673-3690

8. Zhang, L., N. Potter, K. Hickel, Y. Q. Zhang, and Q. X. Shao (2008). "Water balance modeling over variable time scales based on the Budyko framework–Model development and testing." *J. Hydrol.*, 360(1–4), 117–131.

---

## Author Comment (AC2) · 6 Dec 2017

This study estimates the water yield for the Upper Ganga basin using variations of the Budyko model that relates aridity index (ratio of long term potential evapotranspiration to precipitation) to evaporation ratio (ratio of long term actual evapotranspiration to precipitation). Several versions of a parameterized form of the Budyko curve, developed over the past years, are used to estimate long term streamflow for the basin. It is also assessed whether the inclusion of spatial variability of basin properties improves predictive skill. Though the study does not make any new contributions, it has the potential to contribute to understanding hydrology of this particular basin. However, in its present form it has major drawbacks:

1. Literature review: The study overlooks a significant body of literature in streamflow modeling in the region. Studies are available both at the scale of entire India, the Ganga basin as well as finer scales. In addition, the premise of the study is poorly developed and developments related to Budyko's theory are improperly explained. In fact, the work by Donohue et al. (2012) is cited but equations from InVEST's online documentation are instead used. It is not straightforward to connect the equations in the manuscript with Donohue et al. (2012) formulations. Overall, the introduction needs connection to a wider literature base, along with better exposition of developments in Budyko theory.

Reply: The present study focuses on incorporation of spatial variability of various parameters involved in computing water yield using InVEST model. The work does not involve modelling of streamflow rather it attempts to compare the outcomes of spatially distributed water yield model and conventionally used lumped Zhang model. Authors agree that the literature on hydrological modelling of water balance components is available for Ganga basin and its sub-catchments (finer scale), however, the term 'finer scale' in the paper represents incorporation of spatial variations through pixel level estimation of parameters involved in InVEST model which are otherwise taken as lumped. Authors agree that the parameter '$w$' in the equation involved in strategy "E" have been proposed by Donohue et al. (2012) which is also cited in online documentation of InVEST model, however, the final equation used for estimating water yield is from the InVEST model. Considering this fact, Donohue et al. (2012) has been cited in Strategy 'E'. If suggested by reviewer, the citation can be removed from the Strategy 'E'. Various advancements in the Budyko's theory have been addressed properly in revised manuscript.

2. Methods: The methods rely on previously developed relationships between Budyko parameter and observable catchment properties. However, some of these relationships, such as those in Donohue et al. 2012 were developed for Australia. Similarly, Xu et al. (2013) report that the global model could explain only 53% of observed variation of Budyko's parameter in their dataset. The large basin model worked well but is the Upper Ganga basin large enough in comparison to the 32 basins used in Xu et al. (2013)?

Reply: The Donohue et al. (2012) model was developed for Australia, however, the online documentation on InVEST model also states its application globally. Although, the Upper Ganga basin lies in large basin category as per the definition from Xu et al. (2013), but, the yield computed using global model is in good agreement with the observed data for the Upper Ganga basin.

**3. Climate data: The resolution of climate data used to compute fine scale variables is of concern. The introduction stresses on a stronger control of precipitation (and potential evapotranspiration) on runoff estimates, as compared to Budyko's parameter, but the analysis works with coarse climatic data. Though precipitation and temperature data were downscaled to the resolution of land use data (by a statistical technique that is not described well.), the effect of elevation on these variables was neglected (for example lapse rate was not accounted for in temperature estimates). As the basin has significant elevation variations, this may lead to biases in water yield estimates.**

**Reply:** The climate datasets used in the present study is at the finest resolution available so far for the study region. The precipitation and temperature data sets were downscaled to a resolution of land use data using Spline interpolation technique. The details regarding Spline interpolation technique has been added in the revised manuscript. Gridded datasets of temperature and precipitation used in the present study has been developed using quality controlled stations and well-proven interpolation technique. Further details about the datasets are given in Srivastava et al. (2009) and Pai et al. (2014).

**4. Validation: For the validation catchment, 32% of observed discharge is removed as it is assumed to be snowmelt. But snow melt still counts within the hydrological budget of the region as it is contributed by precipitation falling as snow, which is being used in the Budyko model. If the melt contribution was from long term glacier melts that contribute water to the region in addition to precipitation falling as rain or snow, one may remove it. Even that will be challenging at annual time scales if the basin has significant storage. Unless the distinction between glacier and snow melt is made, and some reasoning as to why Budyko's approach can be applied at annual time steps, it will be hard to justify this reduction. There is also the issue of claiming predictive skill over an entire basin by looking at performance at a single sub-basin in a single year. Note that most approaches based on the Budyko's curve must work with long term data as even at annual time scales, catchment's water storage changes may be significant and the Budyko model may be invalid (Donohue et al. 2007). The discussion should reflect the limitations of this approach.**

**Reply:** Present work considers runoff from both precipitation as well as snowfall for the region, but 32% of the observed discharge has been removed as it is contributed by glacier ice melt to the streamflow for this catchment as explained by Maurya et al. (2011) for our study area. The above mentioned fraction of discharge had been quantified using isotope study which separates snow melt contribution from that of the glacier melt (Maurya et al. 2011). The present study attempts to quantify annual water yield at pixel level irrespective of the size of catchment. Therefore, the proposed methodology is expected to perform well for the catchment of any given size. Changes in catchment's water storage over time are required to be quantified in order to validate the applicability of Budyko's model to long term data for the catchment under study. This limitation of the proposed methodology has been added in the revised manuscript.

**5. Interpretation of results: For some reason, as we move from strategy A to E, catchment water yield steadily increases, or, ET decreases. This indicates a systematic change in the Budyko parameter as we go from simpler to more complex relationships requiring more data. Why would the Budyko parameter scale in this manner? This also seems to be in**

**contradiction of the result by Choudhary (1999) who showed that as larger areas are used in a lumped form, Budyo's parameter changes such that actual evapotranspiration reduces. See also the discussion in Donohue et al. (2007). Given the limited data for validation, it is important to physically interpret the results instead of focusing on which method is the best.**

**Reply:** Values of water yield estimated using strategy A to E are systematically increasing but are not steady in nature as water yield estimated using strategy A and B lies in range 650 – 750 mm whereas water yield from strategy C-E lies in range 1229 – 1231 mm for the year 1980 (see Table 1). Similar results are also evident for other years too. Also, water yield estimated using strategy C-E are more or less same for a given year as these strategies involve pixel based estimation of water yield considering spatial variation in Budyko parameters. Parameters involved in Budyko model such as 'w' are found to be dependent on various factors such as catchment characteristics, vegetation cover, etc. as well as climate seasonality (Li et al. 2013). Ahn and Merwade (2017) have analysed the relationship between basin characteristics and factor '*w*' for 175 stations spread over the USA. Results are shown in Fig. 1 (Ahn and Merwade, 2017). As evident from figure, no precise conclusion can be drawn regarding relationship between basin characteristics and value of 'w' especially in case of basin area characteristics. In that case, rationalizing the relationship between basin size and value of Budyko model parameters as documented by Choudhary (1999) is not appropriate. Moreover, no straight forward relationship has yet been identified between basin characteristics and model parameters and it is a subject matter for further study. Authors again want to emphasize over the fact that study focuses on analyzing estimates of water yield computed considering spatial variation in Budyko model parameters at pixel level with water yield computed considering model parameters as lumped for the entire catchment. Authors agree that the data available for validation of parameters estimated at various levels are limited, however, estimated values of AET and PET used in computation of water yield are validated using satellite estimate of the variables for corresponding years (Table 1). From the comparison, both AET (GLDAS) and PET (CRU TS) values are found to in agreement with the satellite estimates. Necessary tables are added to the revised manuscript.

[Figure]

[Figure]

Figure 1: The relationship between basin characteristics and optimal w values (Source: Ahn and Merwade, 2017)

**Table 1: Comparison of model estimated water yield with satellite estimates**

| Parameter | | | | InVEST model | | | | |
|---|---|---|---|---|---|---|---|---|
| (mm) | Year | Source 2 (GLDAS) | Source 2 (CRU) | Strategy A (Lumped Zhang Model) | Strategy B (Large Model) | Strategy C (Global model) | Strategy D (Xu et al. 2013) | Strategy E (Donohue et al. 2012) |
| AET | 1980 | 555.0355 | | 696.84 | 486.07 | 679.52 | 679.68 | 680.01 |
| | 1990 | 646.168 | | 815.02 | 592.3 | 735.23 | 735.27 | 736.25 |
| | 2001 | 588.084 | | 680.76 | 408.86 | 548.28 | 548.39 | 550.38 |
| | 2015 | 716.8316 | | 900.11 | 625.41 | 743.48 | 743.52 | 744.34 |
| | | | | | | | | |
| PET | 1980 | | 1175.964 | 1376.64 | 1382.12 | 1382.12 | 1382.12 | 1382.12 |
| | 1990 | | 1156.497 | 1456.16 | 1461.86 | 1461.86 | 1461.86 | 1461.86 |

| | 2001 | | 1184.847 | 1457.08 | 1462.96 | 1462.96 | 1462.96 | 1462.96 |
| | 2015 | | 1156.686 | 1544.20 | 1550.42 | 1550.42 | 1550.42 | 1550.42 |

**Minor Comment**
**Structure: The paper can be re-structured to improve clarity. Sections 2 and 4 have overlapping**
**items, while 'data' generally goes better with 'Study area'.**

**Reply:** Review suggestions regarding modification of structure of the paper are duly considered in the revised manuscript. Our endeavor will be that the revised paper is much better than the current version.

**References**:

1. Ahn, K. H., and Merwade, V. (2017). "The Integrated Impact of Basin Characteristics on Changes in Hydrological Variables", Book Chapter 12 in "*Sustainable Water Resources Management*", American Society of Civil Engineers (ASCE), pp. 317-336. ISBN: 978-0-7844-1476-7.
2. Choudhury, B. (1999). "Evaluation of an empirical equation for annual evaporation using field observations and results from a biophysical model." *Journal of Hydrology*, 216(1), 99-110.
3. Donohue, R. J., Roderick, M. L., and McVicar, T. R. (2012). "Roots, storms and soil pores: Incorporating key ecohydrological processes into Budyko's hydrological model." *Journal of Hydrology*, 436, 35-50.
4. Li, D., Pan, M., Cong, Z., Zhang, L., and Wood, E. (2013). "Vegetation control on water and energy balance within the Budyko framework." *Water Resources Research*, 49(2), 969-976.
5. Maurya, A. S., Shah, M., Deshpande, R. D., Bhardwaj, R. M., Prasad, A., and Gupta, S. K. (2011). "Hydrograph separation and precipitation source identification using stable water isotopes and conductivity: River Ganga at Himalayan foothills." *Hydrological Processes*, 25(10), 1521-1530.
6. Pai, D. S., Sridhar, L., Rajeevan, M., Sreejith, O. P., Satbhai, N. S., and Mukhopadhyay, B. (2014). "Development of a new high spatial resolution (0.25× 0.25) long period (1901–2010) daily gridded rainfall data set over India and its comparison with existing data sets over the region." *Mausam*, 65(1), 1-18.
7. Srivastava, A. K., Rajeevan, M., and Kshirsagar, S. R. (2009). "Development of a high resolution daily gridded temperature data set (1969–2005) for the Indian region." *Atmospheric Science Letters*, 10(4), 249-254.
8. Xu, X., Liu, W., Scanlon, B. R., Zhang, L., and Pan, M. (2013). "Local and global factors controlling water-energy balances within the Budyko framework." *Geophysical Research Letters*, 40(23), 6123-6129.